# EFFICIENT COMPETITIVE SELF-PLAY POLICY OPTIMIZATION

## ABSTRACT

Reinforcement learning from self-play has recently reported many successes. Self-play, where the agents compete with themselves, is often used to generate training data for iterative policy improvement. In previous work, heuristic rules are designed to choose an opponent for the current learner. Typical rules include choosing the latest agent, the best agent, or a random historical agent. However, these rules may be inefficient in practice and sometimes do not guarantee convergence even in the simplest matrix games. This paper proposes a new algorithmic framework for competitive self-play reinforcement learning in two-player zero-sum games. We recognize the fact that the Nash equilibrium coincides with the saddle point of the stochastic payoff function, which motivates us to borrow ideas from classical saddle point optimization literature. Our method simultaneously trains several agents and intelligently takes each other as opponents based on a simple adversarial rule derived from a principled perturbation-based saddle optimization method. We prove theoretically that our algorithm converges to an approximate equilibrium with high probability in convex-concave games under standard assumptions. Beyond the theory, we further show the empirical superiority of our method over baseline methods relying on the aforementioned opponent-selection heuristics in matrix games, grid-world soccer, Gomoku, and simulated robot sumo, with neural net policy function approximators.

## 1 INTRODUCTION

Reinforcement learning (RL) from self-play has drawn tremendous attention over the past few years. Empirical successes have been observed in several challenging tasks, including Go (Silver et al., 2016; 2017; 2018), simulated hide-and-seek (Baker et al., 2020), simulated sumo wrestling (Bansal et al., 2017), Capture the Flag (Jaderberg et al., 2019), Dota 2 (Berner et al., 2019), StarCraft II (Vinyals et al., 2019), and poker (Brown & Sandholm, 2019), to name a few. During RL from self-play, the learner collects training data by competing with an opponent selected from its past self or an agent population. Self-play presumably creates an auto-curriculum for the agents to learn at their own pace. At each iteration, the learner always faces an opponent that is comparably in strength to itself, allowing continuous improvement.

The way the opponents are selected often follows human-designed heuristic rules in prior work. For example, AlphaGo (Silver et al., 2016) always competes with the latest agent, while the later generation AlphaGo Zero (Silver et al., 2017) and AlphaZero (Silver et al., 2018) generate self-play data with the maintained best historical agent. In specific tasks, such as OpenAI's sumo wrestling, competing against a randomly chosen historical agent leads to the emergence of more diverse behaviors (Bansal et al., 2017) and more stable training than against the latest agent (Al-Shedivat et al., 2018). In population-based training (Jaderberg et al., 2019; Liu et al., 2019) and AlphaStar (Vinyals et al., 2019), an elite or random agent is picked from the agent population as the opponent.

Unfortunately, these rules may be inefficient and sometimes ineffective in practice since they do not necessarily enjoy last-iterate convergence to the "average-case optimal" solution even in tabular matrix games. In fact, in the simple Matching Pennies game, self-play with the latest agent fails to converge and falls into an oscillating behavior, as shown in Sec. 5.

In this paper, we develop an algorithm that adopts a principle-derived opponent-selection rule to alleviate some of the issues mentioned above. This requires clarifying first what the solution of

self-play RL should be. From the game-theoretical perspective, Nash equilibrium is a fundamental solution concept that characterizes the desired "average-case optimal" strategies (policies). When each player assumes other players also play their equilibrium strategies, no one in the game can gain more by unilaterally deviating to another strategy. Nash, in his seminal work (Nash, 1951), has established the existence result of mixed-strategy Nash equilibrium of any finite game. Thus solving for a mixed-strategy Nash equilibrium is a reasonable goal of self-play RL.

We consider the particular case of two-player zero-sum games as the model for the competitive self-play RL environments. In this case, the Nash equilibrium is the same as the (global) saddle point and as the solution of the minimax program $\min_{x \in X} \max_{y \in Y} f(x, y)$. We denote $x, y$ as the strategy profiles (in RL terminology, policies) and $f$ as the loss for $x$ or utility/reward for $y$. A saddle point $(x^*, y^*) \in X \times Y$, where $X, Y$ are the sets of all possible mixed-strategies (stochastic policies) of the two players, satisfies the following key property

$$f(x^*, y) \leq f(x^*, y^*) \leq f(x, y^*), \ \forall x \in X, \forall y \in Y. \tag{1}$$

Connections to the saddle problem and game theory inspire us to borrow ideas from the abundant literature for finding saddle points in the optimization field (Arrow et al., 1958; Korpelevich, 1976; Kallio & Ruszczynski, 1994; Nedić & Ozdaglar, 2009) and for finding equilibrium in the game theory field (Zinkevich et al., 2008; Brown, 1951; Singh et al., 2000). One particular class of method, i.e., the perturbation-based subgradient methods to find the saddles (Korpelevich, 1976; Kallio & Ruszczynski, 1994), is especially appealing. This class of method directly builds upon the inequality properties in Eq. 1, and has several advantages: (1) Unlike some algorithms that require knowledge of the game dynamics (Silver et al., 2016; 2017; Nowé et al., 2012), it requires only subgradients; thus, it is easy to be adapted to policy optimization with estimated policy gradients. (2) For convex-concave functions, it is guaranteed to converge in its last iterate instead of an average iterate, hence alleviates the need to compute any historical averages as in Brown (1951); Singh et al. (2000); Zinkevich et al. (2008), which can get complicated when neural nets are involved (Heinrich & Silver, 2016). (3) Most importantly, it prescribes a simple principled way to adversarially choose self-play opponents, which can be naturally instantiated with a concurrently-trained agent population.

To summarize, we apply ideas from the perturbation-based methods of classical saddle point optimization to the model-free self-play RL regime. This results in a novel population-based policy gradient method with a principled adversarial opponent-selection rule. Analogous to the standard model-free RL setting, we assume only "naive" players (Jafari et al., 2001) where the game dynamic is hidden and only rewards for their own actions are revealed. This enables broader applicability to problems with mismatched or unknown game dynamics than many existing algorithms (Silver et al., 2016; 2017; Nowé et al., 2012). In Sec. 4, we provide an approximate convergence theorem for convex-concave games as a sanity check. Sec. 5 shows extensive experiment results favoring our algorithm's effectiveness in several games, including matrix games, grid-world soccer, a board game, and a challenging simulated robot sumo game. Our method demonstrates higher per-agent sample efficiency than baseline methods with alternative opponent-selection rules. Our trained agents also outperform the baseline agents on average in competitions.

## 2 RELATED WORK

Reinforcement learning trains a single agent to maximize the expected return in an environment (Sutton & Barto, 2018). Multiagent reinforcement learning (MARL), of which two-agent is a special case, concerns multiple agents taking actions in the same environment (Littman, 1994). Self-play is a training paradigm to generate data for MARL and has led to great successes, achieving super-human performance in several domains (Tesauro, 1995; Silver et al., 2016; Brown & Sandholm, 2019). Applying RL algorithms naively as independent learners in MARL sometimes produces strong agents (Tesauro, 1995) but not always. People have studied ways to extend RL algorithms specifically to MARL, e.g., minimax-Q (Littman, 1994), Nash-Q (Hu & Wellman, 2003), WoLF-PG (Bowling & Veloso, 2002), etc. However, most of these methods are designed for tabular RL only, therefore not readily applicable to continuous state action spaces or complex policy functions where gradient-based policy optimization methods are preferred. Recently, Bai & Jin (2020), Lee et al. (2020) and Zhang et al. (2020) provide theoretical regret or convergence analyses under tabular or other restricted self-play settings, which complement our empirical effort.

There are algorithms developed from the game theory and online learning perspective (Lanctot et al., 2017; Nowé et al., 2012; Cardoso et al., 2019), notably Tree search, Fictitious self-play (Brown,

1951), Regret minimization (Jafari et al., 2001; Zinkevich et al., 2008), and Mirror descent (Mertikopoulos et al., 2019; Rakhlin & Sridharan, 2013). Tree search such as minimax and alpha-beta pruning is particularly effective in small-state games. Monte Carlo Tree Search (MCTS) is also effective in Go (Silver et al., 2016). However, Tree search requires learners to know (or at least learn) the game dynamics. The other ones typically require maintaining some historical quantities. In Fictitious play, the learner best-responds to a historical average opponent, and the average strategy converges. Similarly, the total historical regrets in all (information) states are maintained in (counterfactual) regret minimization (Zinkevich et al., 2008). Furthermore, most of those algorithms are designed only for discrete state action games. Special care has to be taken with neural net function approximators (Heinrich & Silver, 2016). On the contrary, our method does not require the complicated computation of averaging neural nets, and is readily applicable to continuous environments.

In two-player zero-sum games, the Nash equilibrium coincides with the saddle point. This enables the techniques developed for finding saddle points. While some saddle-point methods also rely on time averages (Nedić & Ozdaglar, 2009), a class of perturbation-based gradient method is known to converge under mild convex-concave assumption for deterministic functions (Kallio & Ruszczynski, 1994; Korpelevich, 1976; Hamm & Noh, 2018). We develop a sampling version of them for stochastic RL objectives, which leads to a more principled and effective way of choosing opponents in self-play. Our adversarial opponent-selection rule bears a resemblance to Gleave et al. (2019). However, our goal is to develop an effective self-play RL algorithm, while Gleave et al. (2019) aims at attacking deep self-play learned policies. A recent work by Prajapat et al. (2020) tackles the self-play policy optimization problem differently from ours by employing a bilinear approximation to the game. Finally, although the algorithm presented here builds upon policy gradient, the same framework may be extended to other RL algorithms such as MCTS thanks to a recent interpretation of MCTS as policy optimization (Grill et al., 2020). Our way of leveraging Eq. 1 in a population may potentially work beyond gradient-based RL, e.g., in training generative adversarial networks similarly to Hamm & Noh (2018) due to the same minimax formulation.

## 3 METHOD

Classical game theory defines a two-player zero-sum game as a tuple $(X, Y, f)$ where $X, Y$ are the sets of possible strategies of Players 1 and 2 respectively, and $f : X \times Y \mapsto \mathbb{R}$ is a mapping from a pair of strategies to a real-valued utility/reward for Player 2. The game is zero-sum (fully competitive), so Player 1's reward is $-f$. This is a special case of the Stochastic Game formulation for Multiagent RL (Shapley, 1953) which itself is an extension to Markov Decision Processes (MDP).

We consider mixed strategies induced by stochastic policies $\pi_x$ and $\pi_y$. The policies can be parameterized functions in which case $X, Y$ are the sets of all possible policy parameters. Denote $a_t$ as the action of Player 1 and $b_t$ as the action of Player 2 at time $t$, let $T$ be the time limit of the game, then the stochastic payoff $f$ writes as

$$f(x, y) = \mathbb{E}_{\substack{a_t \sim \pi_x, b_t \sim \pi_y, \\ s_{t+1} \sim P(\cdot|s_t, a_t, b_t)}} \left[ \sum_{t=0}^{T} \gamma^t r(s_t, a_t, b_t) \right]. \tag{2}$$

The state sequence $\{s_t\}_{t=0}^{T}$ follows a transition dynamic $P(s_{t+1}|s_t, a_t, b_t)$. Actions are sampled according to action distributions $\pi_x(\cdot|s_t)$ and $\pi_y(\cdot|s_t)$. And $r(s_t, a_t, b_t)$ is the reward (payoff) for Player 2 at time $t$, determined jointly by the state and actions. We use the term 'agent' and 'player' interchangeably. While we consider an agent pair $(x, y)$ in this paper, in some cases (Silver et al., 2016), $x = y$ can be enforced by sharing parameters if the game is impartial. The discounting factor $\gamma$ weights between short- and long-term rewards and is optional.

Note that when one agent is fixed, taking $y$ as an example, the problem $x$ is facing reduces to an MDP if we define a new state transition dynamic $P_{\text{new}}(s_{t+1}|s_t, a_t) = \sum_{b_t} P(s_{t+1}|s_t, a_t, b_t)\pi_y(b_t|s_t)$ and a new reward $r_{\text{new}}(s_t, a_t) = \sum_{b_t} r(s_t, a_t, b_t)\pi_y(b_t|s_t)$. This leads to the naive gradient descent-ascent algorithm, which provably works in strictly convex-concave games (where $f$ is strictly convex in $x$ and strictly concave in $y$) under some assumptions (Arrow et al., 1958). However, in general, it does not enjoy last-iterate convergence to the Nash equilibrium. Even for simple games such as Matching Pennies and Rock Paper Scissors, as we shall see in our experiments, the naive algorithm generates cyclic sequences of $x^k, y^k$ that orbit around the equilibrium. This motivates us to study the perturbation-based method which converges under weaker assumptions.

---

**Algorithm 1:** Perturbation-based self-play policy optimization of an $n$ agent population.

---

**Input:** $N$: Nº iterations; $\eta_k$: learning rates; $m_k$: sample size; $n$: population size; $l$: Nº inner updates;
**Result:** $n$ pairs of policies;

1  Initialize $(x_i^0, y_i^0), i = 1, 2, \ldots n$;
2  **for** $k = 0, 1, 2, \ldots N - 1$ **do**
3      Evaluate $\hat{f}(x_i^k, y_j^k), \forall i, j \in 1 \ldots n$ with Eq. 4 and sample size $m_k$;
4      **for** $i = 1, \ldots n$ **do**
5          Construct candidate opponent sets $C_{y_i}^k = \{y_j^k : j = 1 \ldots n\}$ and $C_{x_i}^k = \{x_j^k : j = 1 \ldots n\}$;
6          Find perturbed $v_i^k = \arg\max_{y \in C_{y_i}^k} \hat{f}(x_i^k, y)$, perturbed $u_i^k = \arg\min_{x \in C_{x_i}^k} \hat{f}(x, y_i^k)$;
7          Invoke a single-agent RL algorithm (e.g., A2C, PPO) on $x_i^k$ for $l$ times that:
8              Estimate policy gradients $\hat{g}_{x_i}^k = \hat{\nabla}_x f(x_i^k, v_i^k)$ with sample size $m_k$ (e.g., Eq. 5);
9              Update policy by $x_i^{k+1} \leftarrow x_i^k - \eta_k \hat{g}_{x_i}^k$ (or RmsProp);
10         Invoke a single-agent RL algorithm (e.g., A2C, PPO) on $y_i^k$ for $l$ times that:
11             Estimate policy gradients $\hat{g}_{y_i}^k = \hat{\nabla}_y f(u_i^k, y_i^k)$ with sample size $m_k$;
12             Update policy by $y_i^{k+1} \leftarrow y_i^k + \eta_k \hat{g}_{y_i}^k$ (or RmsProp);
13 **return** $\{(x_i^N, y_i^N)\}_{i=1}^n$;

---

Recall that the Nash equilibrium has to satisfy the saddle constraints Eq. 1: $f(x^*, y) \leq f(x^*, y^*) \leq f(x, y^*)$. The perturbation-based methods build upon this property (Nedić & Ozdaglar, 2009; Kallio & Ruszczynski, 1994; Korpelevich, 1976) and directly optimize for a solution that meets the constraints. They find perturbed points $u$ of Player 1 and $v$ of Player 2, and use gradients at $(x, v)$ and $(u, y)$ to optimize $x$ and $y$ respectively. Under some regularity assumptions, gradient direction from a single perturbed point is adequate for proving convergence for (not strictly) convex-concave functions (Nedić & Ozdaglar, 2009). They can be easily extended to accommodate gradient based policy optimization and the stochastic RL objective in Eq. 4.

We propose to find the perturbations from an agent population, resulting in the algorithm outlined in Alg. 1. The algorithm trains $n$ pairs of agents simultaneously. At each rounds of training, we first run $n^2$ pairwise competitions as the evaluation step (Alg. 1 L3), costing $n^2 m_k$ trajectories. To save sample complexity, we can use these rollouts to do one policy update as well. Then a simple adversarial rule (Eq. 3) is adopted in Alg. 1 L6 to choose the opponents adaptively. The intuition is that $v_i^k$ and $u_i^k$ are the most challenging opponents in the population for the current $x_i$ and $y_i$.

$$v_i^k = \arg\max_{y \in C_{y_i}^k} \hat{f}(x_i^k, y), \quad u_i^k = \arg\min_{x \in C_{x_i}^k} \hat{f}(x, y_i^k). \tag{3}$$

The perturbations $v_i^k$ and $u_i^k$ always satisfy $f(x_i^k, v_i^k) \geq f(u_i^k, y_i^k)$, since $\max_{y \in C_{y_i}^k} \hat{f}(x_i^k, y) \geq \hat{f}(x_i^k, y_i^k) \geq \min_{x \in C_{x_i}^k} \hat{f}(x, y_i^k)$. Then we run gradient descent on $x_i^k$ with the perturbed $v_i^k$ as opponent to minimize $f(x_i^k, v_i^k)$, and run gradient ascent on $y_i^k$ to maximize $f(u_i^k, y_i^k)$. Intuitively, the duality gap between $\min_x \max_y f(x, y)$ and $\max_y \min_x f(x, y)$, approximated by $f(x_i^k, v_i^k) - f(u_i^k, y_i^k)$, is reduced, leading $(x_i^k, y_i^k)$ to converge to the saddle point (equilibrium).

We build the candidate opponent sets in L5 of Alg. 1 simply as the concurrently-trained $n$-agent population. Specifically, $C_{y_i}^k = \{y_1^k, \ldots, y_n^k\}$ and $C_{x_i}^k = \{x_1^k, \ldots, x_n^k\}$. This is due to the following considerations. An alternative source of candidates is the fixed known agents such as a rule-based agent, which may not be available in practice. Another source is the extragradient methods (Korpelevich, 1976; Mertikopoulos et al., 2019), where extra gradient steps are taken on $y$ before optimizing $x$. The extragradient method can be thought of as a local approximation to Eq. 3 with a neighborhood opponent set, thus is related to our method. However, this method could be less efficient because the trajectory sample used in the extragradient steps is wasted as it does not contribute to actually optimizing $y$. Yet another source is the past agents. This choice is motivated by Fictitious play and ensures that the current learner always defeats a past self. However, as we shall see in the experiments, self-play with a random past agent may learn slower than our method. We expect all agents in the population in our algorithm to be strong, thus provide stronger learning signals.

Finally, we use Monte Carlo estimation to compute the values and gradients of $f$. In the classical game theory setting where the game dynamic and payoff are known, it is possible to compute the exact values and gradients of $f$. But in the model-free MARL setting, we have to collect roll-out trajectories to estimate both the function values through policy evaluation and gradients through

the Policy gradient theorem (Sutton & Barto, 2018). After collecting $m$ independent trajectories $\left\{ \{(s_t^i, a_t^i, r_t^i)\}_{t=0}^T \right\}_{i=1}^m$, we can estimate $f(x, y)$ by

$$\hat{f}(x, y) = \frac{1}{m} \sum_{i=1}^m \sum_{t=0}^T \gamma^t r_t^i. \tag{4}$$

And given estimates $\hat{Q}_x(s, a; y)$ to the state-action value $Q_x(s, a; y)$ (assuming an MDP with $y$ as a fixed opponent of $x$), we construct an estimator for $\nabla_x f(x, y)$ (and similarly for $\nabla_y f$ given $\hat{Q}_y$) by

$$\hat{\nabla}_x f(x, y) \propto \frac{1}{m} \sum_{i=1}^m \sum_{t=0}^T \nabla_x \log \pi_x(a_t^i | s_t^i) \hat{Q}_x(s_t^i, a_t^i; y). \tag{5}$$

## 4 CONVERGENCE ANALYSIS

We establish an asymptotic convergence result in the Monte Carlo policy gradient setting in Thm. 2 for a variant of Alg. 1 under regularity assumptions. This variant sets $l = 1$ and uses the vanilla SGD as the policy optimizer. We add a stop criterion $\hat{f}(x_i^k, v^k) - \hat{f}(u^k, y_i^k) \lesssim \epsilon$ after Line 6 with an accuracy parameter $\epsilon$. The full proof can be found in the appendix. Since the algorithm is symmetric between different pairs of agents in the population, we drop the subscript $i$ for text clarity.

**Assumption 1** (A1). *$X, Y \subseteq \mathbb{R}^d$ are compact sets. As a consequence, there exists $D$ s.t $\forall x_1, x_2 \in X$, $\|x_1 - x_2\|_1 \leq D$ and $\forall y_1, y_2 \in Y$, $\|y_1 - y_2\|_1 \leq D$. Assume $C_y^k, C_x^k$ are compact subsets of $X$ and $Y$. Further, assume $f: X \times Y \mapsto \mathbb{R}$ is a bounded convex-concave function.*

**Theorem 1** (Convergence with exact gradients (Kallio & Ruszczynski, 1994)). *Under A1, if a sequence $(x^k, y^k) \to (\hat{x}, \hat{y}) \wedge f(x^k, v^k) - f(u^k, y^k) \to 0$ implies $(\hat{x}, \hat{y})$ is a saddle point, Alg. 1 (replacing estimates with true values) produces a sequence $\left\{ (x^k, y^k) \right\}_{k=0}^{\infty}$ convergent to a saddle.*

The above case with exact sub-gradients is easy since both $f$ and $\nabla f$ are deterministic. In RL setting, we construct estimates for $f(x, y)$ and $\nabla_x f, \nabla_y f$ with samples. Intuitively, when the samples are large enough, we can bound the deviation between the true values and estimates by concentration inequalities, then the proof outline similar to Kallio & Ruszczynski (1994) also goes through.

Thm. 2 requires an extra assumption on the boundedness of $\hat{Q}$ and gradients. By showing the policy gradient estimates are approximate sub-/super-gradients of $f$, we are able to prove that the output $(x_i^N, y_i^N)$ of Alg. 1 is an approximate Nash equilibrium with high probability.

**Assumption 2** (A2). *The $Q$ value estimation $\hat{Q}$ is unbiased and bounded by $R$, and the policy has bounded gradient $\|\nabla \log \pi_\theta(a|s)\|_\infty \leq B$.*

**Theorem 2** (Convergence with policy gradients). *Under A1, A2, let sample size at step $k$ be $m_k \geq \Omega\left(\frac{R^2 B^2 D^2}{\epsilon^2} \log \frac{d}{\delta 2^{-k}}\right)$ and learning rate $\eta_k = \alpha \frac{\hat{E}_k - 2\epsilon}{\|\hat{g}_x^k\|^2 + \|\hat{g}_y^k\|^2}$ with $0 \leq \alpha \leq 2$, then with probability at least $1 - \mathcal{O}(\delta)$, the Monte Carlo version of Alg. 1 generates a sequence of points $\left\{ (x^k, y^k) \right\}_{k=0}^{\infty}$ convergent to an $\mathcal{O}(\epsilon)$-approximate equilibrium $(\bar{x}, \bar{y})$, that is $\forall x \in X, \forall y \in Y, f(x, \bar{y}) - \mathcal{O}(\epsilon) \leq f(\bar{x}, \bar{y}) \leq f(\bar{x}, y) + \mathcal{O}(\epsilon)$.*

**Discussion.** The theorems require $f$ to be convex in $x$ and concave in $y$, but not strictly, which is a weaker assumption than Arrow et al. (1958). The purpose of this simple analysis is mainly a sanity check for correctness. It applies to the setting in Sec. 5.1 but not beyond, as the assumptions do not necessarily hold for neural networks. The sample size is chosen loosely as we are not aiming at a sharp finite sample complexity analysis. In practice, we can find suitable $m_k$ (sample size) and $\eta_k$ (learning rates) by experimentation, and adopt a modern RL algorithm with an advanced optimizer (e.g., PPO (Schulman et al., 2017) with RmsProp (Hinton et al.)) in place of the SGD updates.

## 5 EXPERIMENTS

We empirically evaluate our algorithm in several games with distinct characteristics.

**Compared methods.** In Matrix games, we compare to a naive mirror descent method, which is essentially Self-play with the latest agent, to verify convergence. In the rest of the environments, we compare the results from the following methods:

1. **Self-play with the latest agent (Naive Mirror Descent).** The learner always competes with the most recent agent. This is essentially the Gradient Descent Ascent method by Arrow-Hurwicz-Uzawa (Arrow et al., 1958) or the naive mirror/alternating descent.
2. **Self-play with the best past agent.** The learner competes with the best historical agent maintained. The new agent replaces the maintained agent if it beats the existing one. This is the scheme in AlphaGo Zero and AlphaZero (Silver et al., 2017; 2018).
3. **Self-play with a random past agent (Fictitious play).** The learner competes against a randomly sampled historical opponent. This is the scheme in OpenAI sumo (Bansal et al., 2017; Al-Shedivat et al., 2018). It is similar to Fictitious play (Brown, 1951) since uniformly random sampling is equivalent to historical average. However, Fictitious play only guarantees convergence of the average-iterate but not the last-iterate agent.
4. **OURS($n = 2, 4, 6, \ldots$).** This is our algorithm with a population of $n$ pairs of agents trained simultaneously, with each other as candidate opponents. Implementation can be distributed.

**Evaluation protocols.** We mainly measure the strength of agents by the Elo scores (Elo, 1978). Pairwise competition results are gathered from a large tournament among *all* the checkpoint agents of all methods after training. Each competition has multiple matches to account for randomness. The Elo scores are computed by logistic regression, as Elo assumes a logistic relationship $P(\text{A wins}) + 0.5P(\text{draw}) = 1/(1 + 10^{(R_B - R_A)/400})$. A 100 Elo difference corresponds to roughly 64% win-rate. The initial agent's Elo is calibrated to 0. Another way to measure the strength is to compute the average rewards (win-rates) against other agents. We also report average rewards in the appendix.

## 5.1 MATRIX GAMES

We verified the last-iterate convergence to Nash equilibrium in several classical two-player zero-sum matrix games. In comparison, the vanilla mirror descent/ascent is known to produce oscillating behaviors (Mertikopoulos et al., 2019). Payoff matrices (for both players separated by comma), phase portraits, error curves, and our observations are shown in Tab. 1,2,3,4 and Fig. 1,2,3,4.

We studied two settings: (1) OURS(Exact Gradient), the full information setting, where the players know the payoff matrix and compute the exact gradients on action probabilities; (2) OURS(Policy Gradient), the reinforcement learning or bandit setting, where each player only receives the reward of its own action. The action probabilities were modeled by a probability vector $p \in \Delta_2$. We estimated the gradient w.r.t $p$ with REINFORCE estimator (Williams, 1992) with sample size $m_k = 1024$, and applied $\eta_k = 0.03$ constant learning rate SGD with proximal projection onto $\Delta_2$. We trained $n = 4$ agents jointly for Alg. 1 and separately for the naive mirror descent under the same initialization.

## 5.2 GRID-WORLD SOCCER GAME

We conducted experiments in a grid-world soccer game. Similar games were adopted in Littman (1994) and He et al. (2016). Two players compete in a $6 \times 9$ grid world, starting from random positions. The action space is $\{\text{up}, \text{down}, \text{left}, \text{right}, \text{noop}\}$. Once a player scores a goal, it gets positive reward 1.0, and the game ends. Up to $T = 100$ timesteps are allowed. The game ends with a draw if time runs out. The game has imperfect information, as the two players move simultaneously.

The policy and value functions were parameterized by simple one-layer networks, consisting of a one-hot encoding layer and a linear layer that outputs the action logits and values. The logits are transformed into probabilities via $\text{softmax}$. We used Advantage Actor-Critic (A2C) (Mnih et al., 2016) with Generalized Advantage Estimation (Schulman et al., 2016) and RmsProp (Hinton et al.) as the base RL algorithm. The hyper-parameters were $N = 50$, $l = 10$, $m_k = 32$ for Alg. 1. We kept track of the per-agent number of trajectories (episodes) each algorithm used for fair comparison. Other hyper-parameters are listed in the appendix. All methods were run multiple times to calculate the confidence intervals.

In Fig. 5, OURS($n = 2, 4, 6$) all perform better than others, achieving higher Elo scores after experiencing the same number of per-agent episodes. Other methods fail to beat the rule-based agent after 32000 episodes. Competing with a random past agent learns the slowest, suggesting that, though it may stabilize training and diversify behaviors (Bansal et al., 2017), the learning efficiency is not high because a large portion of samples is devoted to weak opponents. Within our method, the performance increases with a larger $n$, suggesting a larger population may help find better perturbations.

## 5.3 GOMOKU BOARD GAME

We investigated the effectiveness in the Gomoku game, which is also known as Renju, Five-in-a-row. In our variant, two players place black or white stones on a 9-by-9 board in turn. The player who

| **Game payoff matrix** | **Phase portraits and error curves** |

|  | Heads | Tails |
|---|---|---|
| Heads | $1, -1$ | $-1, 1$ |
| Tails | $-1, 1$ | $1, -1$ |

Tab. 1: Matching Pennies, a classical game where two players simultaneously turn their pennies to heads or tails. If the pennies match, Player 2 (Row) wins one penny from Player 1 (Column); otherwise, Player 1 wins. $\left(P_x(\text{head}), P_y(\text{head})\right) = \left(\frac{1}{2}, \frac{1}{2}\right)$ is the unique Nash equilibrium with game value 0.

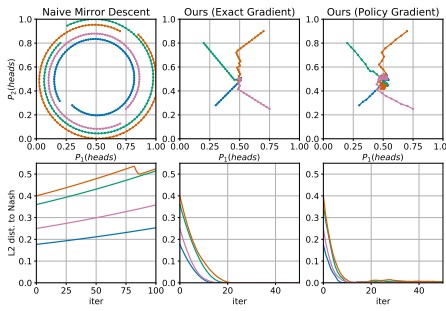

Fig. 1: Matching Pennies. (Top) The phase portraits. (Bottom) The squared $L_2$ distance to the equilibrium. Four colors correspond to the 4 agents in the population with 4 initial points.

|  | Heads | Tails |
|---|---|---|
| Heads | $2, -2$ | $0, 0$ |
| Tails | $-1, 1$ | $2, -2$ |

Tab. 2: Skewed Matching Pennies.

**Observation:** In the leftmost column of Fig. 1,2, the naive mirror descent does not converge pointwisely; Instead, it is trapped in a cyclic behavior. The trajectories of the probability of playing Heads orbit around the Nash, showing as circles in the phase portrait. On the other hand, our method enjoys approximate last-iterate convergence with both exact and policy gradients.

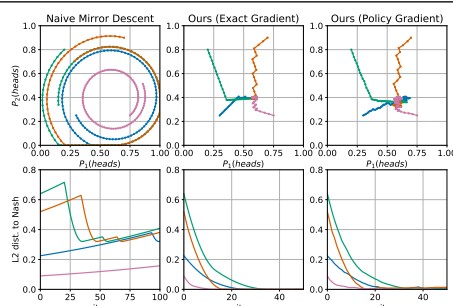

Fig. 2: Skewed Matching Pennies. The unique Nash equilibrium is $\left(P_x(\text{heads}), P_y(\text{heads})\right) = \left(\frac{3}{5}, \frac{2}{5}\right)$ with value 0.8.

|  | Rock | Paper | Scissors |
|---|---|---|---|
| Rock | $0, 0$ | $-1, 1$ | $1, -1$ |
| Paper | $1, -1$ | $0, 0$ | $-1, 1$ |
| Scissors | $-1, 1$ | $1, -1$ | $0, 0$ |

Tab. 3: Rock Paper Scissors.

**Observation:** Similar observations occur in the Rock Paper Scissors game (Fig. 3). The naive method circles around the corresponding equilibrium points $\left(\frac{3}{5}, \frac{2}{5}\right)$ and $\left(\frac{1}{3}, \frac{1}{3}, \frac{1}{3}\right)$, while our method converges with diminishing error.

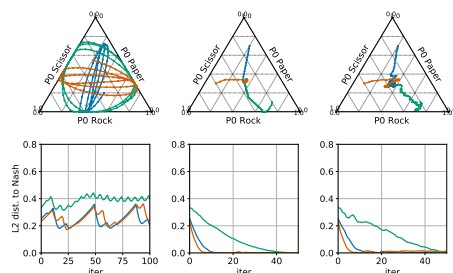

Fig. 3: Rock Paper Scissors. (Top) Visualization of Player 1's strategies ($y_0$) of one of the agents in the population. (Down) The squared distance to equilibrium.

|  | a | b | c |
|---|---|---|---|
| A | $1, -1$ | $-1, 1$ | $0.5, -0.5$ |
| B | $-1, 1$ | $1, -1$ | $-0.5, 0.5$ |

Tab. 4: Extended Matching Pennies.

**Observation:** Our method has the benefit of producing diverse solutions when there exist multiple Nash equilibria. The solution for row player is $x = \left(\frac{1}{2}, \frac{1}{2}\right)$, while any interpolation between $\left(\frac{1}{2}, \frac{1}{2}, 0\right)$ and $\left(0, \frac{1}{3}, \frac{2}{3}\right)$ is an equilibrium column strategy. Depending on initialization, agents in our method converges to different equilibria.

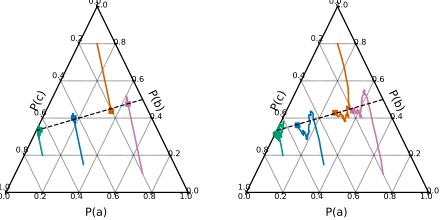

Fig. 4: Visualization of the row player's strategies. (Left) Exact gradient; (Right) Policy gradient. The dashed line represents possible equilibrium strategies. The four agents (in different colors) in the population trained by our algorithm ($n = 4$) converge differently.

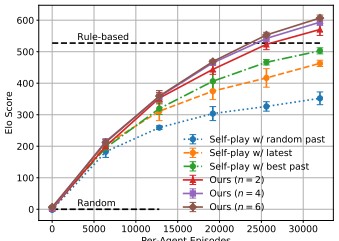 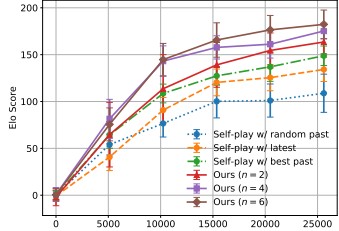 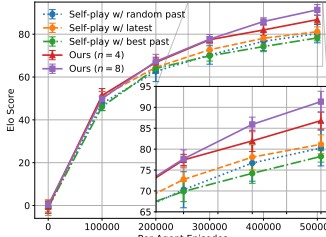

Fig. 5: Soccer Elo curves averaged over 3 runs (random seeds). For OURS(n), the average is over 3n agents. Horizontal lines show the scores of the rule-based and the random-action agents.

Fig. 6: Gomoku Elo curves averaged over 10 runs for the baseline methods, 6 runs (12 agents) for OURS(n = 2), 4 runs (16 agents) for OURS(n = 4), and 3 runs (18 agents) for OURS(n = 6).

Fig. 7: RoboSumo Ants Elo curves averaged over 4 runs for the baseline methods, 2 runs for OURS(n = 4, 8). A close-up is also drawn for better viewing.

\* In all three figures, bars show the 95% confidence intervals. We compare *per-agent* sample efficiency.

gets an unbroken row of five horizontally, vertically, or diagonally, wins (reward 1). The game is a draw (reward 0) when no valid move remains. The game is sequential and has perfect information.

This experiment involved much more complex neural networks than before. We adopted a 4-layer convolutional ReLU network (kernels $(5, 5, 3, 1)$, channels $(16, 32, 64, 1)$, all strides 1) for both the policy and value networks. Gomoku is hard to train from scratch with pure model-free RL without explicit tree search. Hence, we pre-trained the policy nets on expert data collected from `renjuoffline.com`. We downloaded roughly 130 thousand games and applied behavior cloning. The pre-trained networks were able to predict expert moves with $\approx 41\%$ accuracy and achieve an average score of 0.93 (96% win and 4% lose) against a random-action player. We adopted the A2C (Mnih et al., 2016) with GAE (Schulman et al., 2016) and RmsProp (Hinton et al.) with learning rate $\eta_k = 0.001$. Up to $N = 40$ iterations of Alg. 1 were run. The other hyperparameters were the same as those in the soccer game.

In Fig. 6, all methods are able to improve upon the behavior cloning policies significantly. OURS($n = 2, 4, 6$) demonstrate higher sample efficiency by achieving higher Elo ratings than the alternatives given the same amount of per-agent experience. This again suggests that the opponents are chosen more wisely, resulting in better policy improvements. Lastly, the more complex policy and value functions (multi-layer CNN) do not seem to undermine the advantage of our approach.

## 5.4 ROBOSUMO ANTS

Our last experiment is based on the RoboSumo simulation environment in Al-Shedivat et al. (2018) and Bansal et al. (2017), where two Ants wrestle in an arena. This setting is particularly relevant to practical robotics research, as we believe success in this simulation could be transferred into the real-world. The Ants move simultaneously, trying to force the opponent out of the arena or onto the floor. The physics simulator is MuJoCo (Todorov et al., 2012). The observation space and action space are continuous. This game is challenging since it involves a complex continuous control problem with sparse rewards. Following Al-Shedivat et al. (2018) and Bansal et al. (2017), we utilized PPO (Schulman et al., 2017) with GAE (Schulman et al., 2016) as the base RL algorithm, and used a 2-layer fully connected network with width 64 for function approximation. Hyper-parameters $N = 50$, $m_k = 500$. In Al-Shedivat et al. (2018), a random past opponent is sampled in self-play, corresponding to the "Self-play w/ random past" baseline here. The agents are initialized from imitating the pre-trained agents of Al-Shedivat et al. (2018). We considered $n = 4$ and $n = 8$ for our method. From Fig. 7, we observe again that OURS($n = 4, 8$) outperform the baseline methods by a statistical margin and that our method benefits from a larger population size.

## 6 CONCLUSION

We propose a new algorithmic framework for competitive self-play policy optimization inspired by a perturbation subgradient method for saddle points. Our algorithm provably converges in convex-concave games and achieves better per-agent sample efficiency in several experiments. In the future, we hope to study a larger population size (should we have sufficient computing power) and the possibilities of model-based and off-policy self-play RL under our framework.

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

## A EXPERIMENT DETAILS

### A.1 ILLUSTRATIONS OF THE GAMES IN THE EXPERIMENTS

| Illustration | | Properties |
|---|---|---|
| 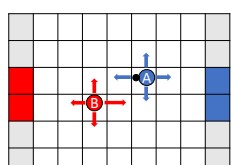 | Fig. 8: Illustration of the 6x9 grid-world soccer game. Red and blue represent the two teams A and B. At start, the players are initialized to random positions on respective sides, and the ball is randomly assigned to one team. Players move up, down, left and right. Once a player scores a goal, the corresponding team wins and the game ends. One player can intercept the other's ball by crossing the other player. | **Observation space**: Tensor of shape $[5,]$, $(x_A, y_A, x_B, y_B,$ A has ball) **Action space**: $\{$up, down, left, right, noop$\}$ **Time limit**: 50 moves **Terminal reward**: $+1$ for winning team $-1$ for losing team $0$ if timeout |
| 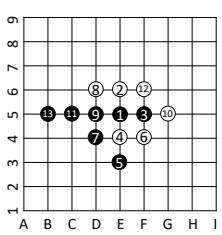 | Fig. 9: Illustration of the Gomoku game (also known as Renju, five-in-a-row). We study the 9x9 board variant. Two players sequentially place black and white stones on the board. Black goes first. A player wins when he or she gets five stones in a row. In the case of this illustration, the black wins because there is five consecutive black stones in the 5th row. Numbers in the stones indicate the ordered they are placed. | **Observation space**: Tensor of shape $[9, 9, 3]$, last dim 0: vacant, 1: black, 2: white **Action space**: Any valid location on the 9x9 board **Time limit**: 41 moves per-player **Terminal reward**: $+1$ for winning player $-1$ for losing player $0$ if timeout |
| | Fig. 10: Illustration of the RoboSumo Ants game. Two ants fight in the arena. The goal is to push the opponent out of the arena or down to the floor. Agent positions are initialized to be random at the start of the game. The game ends in a draw if the time limit is reached. In addition to the terminal $\pm 1$ reward, the environment comes with shaping rewards (motion bonus, closeness to opponent, etc.). In order to make the game zero-sum, we take the difference between the original rewards of the two ants. | **Observation space**: $R^{120}$ **Action space**: $R^8$ **Time limit**: 100 moves **Reward**: $r_t = r_t^{\text{orig y}} - r_t^{\text{orig x}}$ Terminal $\pm 1$ or 0. |

### A.2 HYPER-PARAMETERS

The hyper-parameters in different games are listed in Tab. 5.

### A.3 ADDITIONAL RESULTS

**Win-rates (or average rewards).** Here we report additional results in terms of the average win-rates, or equivalently the average rewards through the linear transform win-rate $= 0.5 + 0.5$ reward, in Tab. 6 and 7. Since we treat each $(x_i, y_i)$ pair as one agent, the values are the average of $f(x_i, \cdot)$ and $f(\cdot, y_i)$ in the first table. The one-side $f(\cdot, y_i)$ win-rates are in the second table. Mean and 95% confidence intervals are estimated from multiple runs. Exact numbers of runs are in the captions

Tab. 5: Hyper-parameters.

| Hyper-param \ Game | Soccer | Gomoku | RoboSumo |
|---|---|---|---|
| Num. of iterations $N$ | 50 | 40 | 50 |
| Learning rate $\eta_k$ | 0.1 | $0 \rightarrow 0.001$ in first 20 steps then 0.001 | 3e-5 $\rightarrow$ 0 linearly |
| Value func learning rate | (Same as above.) | (Same as above.) | 9e-5 |
| Sample size $m_k$ | 32 | 32 | 500 |
| Num. of inner updates $l$ | 10 | 10 | 10 |
| Env. time limit | 50 | 41 per-player | 100 |
| Base RL algorithm | A2C | A2C | PPO, clip 0.2, mini-batch 512, epochs 3 |
| Optimizer | RmsProp, $\alpha = 0.99$ | RmsProp, $\alpha = 0.99$ | RmsProp, $\alpha = 0.99$ |
| Max gradient norm | 1.0 | 1.0 | 0.1 |
| GAE $\lambda$ parameter | 0.95 | 0.95 | 0.98 |
| Discounting factor $\gamma$ | 0.97 | 0.97 | 0.995 |
| Entropy bonus coef. | 0.01 | 0.01 | 0 |
| Policy function | Sequential[ OneHot[5832], Linear[5832,5], Softmax, CategoricalDist ] | Sequential[ Conv[c16,k5,p2], ReLU, Conv[c32,k5,p2], ReLU, Conv[c64,k3,p1], ReLU, Conv[c1,k1], Spatial Softmax, CategoricalDist ] | Sequential[ Linear[120,64], TanH, Linear[64,64], TanH, Linear[64,8], TanH, GaussianDist ] Tanh ensures the mean of the Gaussian is between -1 and 1. The density is corrected. |
| Value function | Sequential[ OneHot[5832], Linear[5832,1] ] | Share 3 Conv layers with the policy, but additional heads: global average and Linear[64,1] | Sequential[ Linear[120,64], TanH, Linear[64,64], TanH, Linear[64,1] ] |

of Fig. 5,6,7 of the main paper. The message is the same as that suggested by the Elo scores: Our method consistently produces stronger agents. We hope the win-rates may give better intuition about the relative performance of different methods.

Tab. 6: Average win-rates ($\in [0, 1]$) between the last-iterate (final) agents trained by different algorithms. Last two rows further show the average over other last-iterate agents and all other agents (historical checkpoint) included in the tournament, respectively. Since an agent consists of an $(x, y)$ pair, the win-rate is averaged on $x$ and $y$, i.e., $\text{win(col vs row)} = \frac{f(x^{\text{row}}, y^{\text{col}}) - f(x^{\text{col}}, y^{\text{row}})}{2} \times 0.5 + 0.5$. The lower the better within each column; The higher the better within each row.

(a) Soccer

| Soccer | Self-play latest | Self-play best | Self-play rand | Ours (n=2) | Ours (n=4) | Ours (n=6) |
|---|---|---|---|---|---|---|
| Self-play latest | - | $0.533 \pm 0.044$ | $0.382 \pm 0.082$ | $0.662 \pm 0.054$ | $0.691 \pm 0.029$ | $0.713 \pm 0.032$ |
| Self-play best | $0.467 \pm 0.044$ | - | $0.293 \pm 0.059$ | $0.582 \pm 0.042$ | $0.618 \pm 0.031$ | $0.661 \pm 0.030$ |
| Self-play rand | $0.618 \pm 0.082$ | $0.707 \pm 0.059$ | - | $0.808 \pm 0.039$ | $0.838 \pm 0.028$ | $0.844 \pm 0.043$ |
| Ours (n=2) | $0.338 \pm 0.054$ | $0.418 \pm 0.042$ | $0.192 \pm 0.039$ | - | $0.549 \pm 0.022$ | $0.535 \pm 0.022$ |
| Ours (n=4) | $0.309 \pm 0.029$ | $0.382 \pm 0.031$ | $0.162 \pm 0.028$ | $0.451 \pm 0.022$ | - | $0.495 \pm 0.023$ |
| Ours (n=6) | $0.287 \pm 0.032$ | $0.339 \pm 0.030$ | $0.156 \pm 0.043$ | $0.465 \pm 0.022$ | $0.505 \pm 0.023$ | - |
| Last-iter average | $0.357 \pm 0.028$ | $0.428 \pm 0.028$ | $0.202 \pm 0.023$ | $0.532 \pm 0.023$ | $0.608 \pm 0.018$ | $0.585 \pm 0.022$ |
| Overall average | $0.632 \pm 0.017$ | $0.676 \pm 0.014$ | $0.506 \pm 0.020$ | $0.749 \pm 0.009$ | $0.775 \pm 0.006$ | $0.776 \pm 0.008$ |

(b) Gomoku

| Gomoku | Self-play latest | Self-play best | Self-play rand | Ours (n=2) | Ours (n=4) | Ours (n=6) |
|---|---|---|---|---|---|---|
| Self-play latest | - | $0.523 \pm 0.026$ | $0.462 \pm 0.032$ | $0.551 \pm 0.024$ | $0.571 \pm 0.018$ | $0.576 \pm 0.017$ |
| Self-play best | $0.477 \pm 0.026$ | - | $0.433 \pm 0.031$ | $0.532 \pm 0.024$ | $0.551 \pm 0.018$ | $0.560 \pm 0.020$ |
| Self-play rand | $0.538 \pm 0.032$ | $0.567 \pm 0.031$ | - | $0.599 \pm 0.027$ | $0.588 \pm 0.022$ | $0.638 \pm 0.020$ |
| Ours (n=2) | $0.449 \pm 0.024$ | $0.468 \pm 0.024$ | $0.401 \pm 0.027$ | - | $0.528 \pm 0.015$ | $0.545 \pm 0.017$ |
| Ours (n=4) | $0.429 \pm 0.018$ | $0.449 \pm 0.018$ | $0.412 \pm 0.022$ | $0.472 \pm 0.015$ | - | $0.512 \pm 0.013$ |
| Ours (n=6) | $0.424 \pm 0.017$ | $0.440 \pm 0.020$ | $0.362 \pm 0.020$ | $0.455 \pm 0.017$ | $0.488 \pm 0.013$ | - |
| Last-iter average | $0.455 \pm 0.010$ | $0.479 \pm 0.011$ | $0.407 \pm 0.012$ | $0.509 \pm 0.010$ | $0.537 \pm 0.008$ | $0.560 \pm 0.008$ |
| Overall average | $0.541 \pm 0.004$ | $0.561 \pm 0.004$ | $0.499 \pm 0.005$ | $0.583 \pm 0.004$ | $0.599 \pm 0.003$ | $0.615 \pm 0.003$ |

(c) RoboSumo

| RoboSumo | Self-play latest | Self-play best | Self-play rand | Ours (n=4) | Ours (n=8) |
|---|---|---|---|---|---|
| Self-play latest | - | $0.502 \pm 0.012$ | $0.493 \pm 0.013$ | $0.511 \pm 0.011$ | $0.510 \pm 0.010$ |
| Self-play best | $0.498 \pm 0.012$ | - | $0.506 \pm 0.014$ | $0.514 \pm 0.008$ | $0.512 \pm 0.010$ |
| Self-play rand | $0.507 \pm 0.013$ | $0.494 \pm 0.014$ | - | $0.508 \pm 0.011$ | $0.515 \pm 0.011$ |
| Ours (n=4) | $0.489 \pm 0.011$ | $0.486 \pm 0.008$ | $0.492 \pm 0.011$ | - | $0.516 \pm 0.008$ |
| Ours (n=8) | $0.490 \pm 0.010$ | $0.488 \pm 0.010$ | $0.485 \pm 0.011$ | $0.484 \pm 0.008$ | - |
| Last-iter average | $0.494 \pm 0.006$ | $0.491 \pm 0.005$ | $0.492 \pm 0.006$ | $0.500 \pm 0.005$ | $0.514 \pm 0.005$ |
| Overall average | $0.531 \pm 0.004$ | $0.527 \pm 0.004$ | $0.530 \pm 0.004$ | $0.539 \pm 0.003$ | $0.545 \pm 0.003$ |

**Training time.** Thanks to the easiness of parallelization, the proposed algorithm enjoys good scalability. We can either distribute the $n$ agents into $n$ processes to run concurrently, or make the rollouts parallel. Our implementation took the later approach. In the most time-consuming RoboSumo Ants experiment, with 30 Intel Xeon CPUs, the baseline methods took approximately 2.4h, while Ours (n=4) took 10.83h to train ($\times 4.5$ times), and Ours (n=8) took 20.75h ($\times 8.6$ times). Note that, Ours (n) trains $n$ agents simultaneously. If we train $n$ agents with the baseline methods by repeating the experiment $n$ times, the time would be $2.4n$ hours, which is comparable to Ours (n).

**Chance of selecting the agent itself as opponent.** One big difference between our method and the compared baselines is the ability to select opponents adversarially from the population. Consider the agent pair $(x_i, y_i)$. When training $x_i$, our method finds the strongest opponent (that incurs the largest loss on $x_i$) from the population, whereas the baselines always choose (possibly past versions of) $y_i$. Since the candidate set contains $y_i$, the "fall-back" case is to use $y_i$ as opponent in our method. We report the frequency that $y_i$ is chosen as opponent for $x_i$ (and $x_i$ for $y_i$ likewise). This gives a sense of how often our method falls back to the baseline method. From Tab. 8, we can observe that, as $n$ grows larger, the chance of fall-back is decreased. This is understandable since a larger population means larger candidate sets and a larger chance to find good perturbations.

## B    PROOFS

We adopt the following variant of Alg. 1 in our asymptotic convergence analysis. For clarity, we investigate the learning process of one agent in the population and drop the $i$ index. $C_x^k$ and $C_y^k$ are

Tab. 7: Average *one-sided* win-rates ($\in [0,1]$) between the last-iterate (final) agents trained by different algorithms. The win-rate is one-sided, i.e., $\text{win}(y^{\text{col}} \text{ vs } x^{\text{row}}) = f(x^{\text{row}}, y^{\text{col}}) \times 0.5 + 0.5$. The lower the better within each column; The higher the better within each row.

(a) Soccer

| row $x$ \ col $y$ | Self-play latest | Self-play best | Self-play rand | Ours (n=2) | Ours (n=4) | Ours (n=6) |
|---|---|---|---|---|---|---|
| Self-play latest | $0.536 \pm 0.054$ | $0.564 \pm 0.079$ | $0.378 \pm 0.103$ | $0.674 \pm 0.080$ | $0.728 \pm 0.039$ | $0.733 \pm 0.048$ |
| Self-play best | $0.497 \pm 0.065$ | $0.450 \pm 0.064$ | $0.306 \pm 0.106$ | $0.583 \pm 0.056$ | $0.601 \pm 0.039$ | $0.642 \pm 0.050$ |
| Self-play rand | $0.614 \pm 0.163$ | $0.719 \pm 0.090$ | $0.481 \pm 0.102$ | $0.796 \pm 0.071$ | $0.816 \pm 0.039$ | $0.824 \pm 0.062$ |
| Ours (n=2) | $0.350 \pm 0.051$ | $0.419 \pm 0.057$ | $0.181 \pm 0.049$ | $0.451 \pm 0.037$ | $0.525 \pm 0.031$ | $0.553 \pm 0.034$ |
| Ours (n=4) | $0.346 \pm 0.046$ | $0.365 \pm 0.047$ | $0.140 \pm 0.034$ | $0.427 \pm 0.034$ | $0.491 \pm 0.020$ | $0.494 \pm 0.033$ |
| Ours (n=6) | $0.308 \pm 0.042$ | $0.319 \pm 0.052$ | $0.136 \pm 0.050$ | $0.483 \pm 0.043$ | $0.505 \pm 0.030$ | $0.515 \pm 0.032$ |
| Last-iter average | $0.381 \pm 0.033$ | $0.422 \pm 0.036$ | $0.188 \pm 0.028$ | $0.525 \pm 0.029$ | $0.601 \pm 0.021$ | $0.587 \pm 0.026$ |
| Overall average | $0.654 \pm 0.017$ | $0.665 \pm 0.016$ | $0.502 \pm 0.021$ | $0.745 \pm 0.010$ | $0.771 \pm 0.006$ | $0.775 \pm 0.009$ |

(b) Gomoku

| row $x$ \ col $y$ | Self-play latest | Self-play best | Self-play rand | Ours (n=2) | Ours (n=4) | Ours (n=6) |
|---|---|---|---|---|---|---|
| Self-play latest | $0.481 \pm 0.031$ | $0.540 \pm 0.038$ | $0.488 \pm 0.050$ | $0.594 \pm 0.041$ | $0.571 \pm 0.026$ | $0.586 \pm 0.030$ |
| Self-play best | $0.494 \pm 0.033$ | $0.531 \pm 0.030$ | $0.471 \pm 0.049$ | $0.597 \pm 0.040$ | $0.562 \pm 0.024$ | $0.572 \pm 0.028$ |
| Self-play rand | $0.565 \pm 0.036$ | $0.605 \pm 0.036$ | $0.572 \pm 0.051$ | $0.668 \pm 0.040$ | $0.617 \pm 0.027$ | $0.647 \pm 0.029$ |
| Ours (n=2) | $0.491 \pm 0.031$ | $0.533 \pm 0.033$ | $0.470 \pm 0.040$ | $0.568 \pm 0.035$ | $0.571 \pm 0.022$ | $0.552 \pm 0.025$ |
| Ours (n=4) | $0.428 \pm 0.022$ | $0.461 \pm 0.024$ | $0.440 \pm 0.035$ | $0.515 \pm 0.029$ | $0.491 \pm 0.017$ | $0.503 \pm 0.020$ |
| Ours (n=6) | $0.435 \pm 0.021$ | $0.453 \pm 0.026$ | $0.370 \pm 0.028$ | $0.462 \pm 0.025$ | $0.479 \pm 0.018$ | $0.467 \pm 0.017$ |
| Last-iter average | $0.472 \pm 0.012$ | $0.506 \pm 0.014$ | $0.438 \pm 0.017$ | $0.549 \pm 0.016$ | $0.550 \pm 0.011$ | $0.564 \pm 0.012$ |
| Overall average | $0.548 \pm 0.005$ | $0.585 \pm 0.005$ | $0.536 \pm 0.007$ | $0.631 \pm 0.006$ | $0.608 \pm 0.004$ | $0.617 \pm 0.004$ |

(c) RoboSumo

| row $x$ \ col $y$ | Self-play latest | Self-play best | Self-play rand | Ours (n=4) | Ours (n=8) |
|---|---|---|---|---|---|
| Self-play latest | $0.516 \pm 0.022$ | $0.494 \pm 0.020$ | $0.491 \pm 0.023$ | $0.502 \pm 0.017$ | $0.511 \pm 0.016$ |
| Self-play best | $0.489 \pm 0.018$ | $0.504 \pm 0.023$ | $0.503 \pm 0.022$ | $0.506 \pm 0.014$ | $0.509 \pm 0.014$ |
| Self-play rand | $0.505 \pm 0.021$ | $0.491 \pm 0.026$ | $0.494 \pm 0.026$ | $0.518 \pm 0.017$ | $0.516 \pm 0.014$ |
| Ours (n=4) | $0.480 \pm 0.018$ | $0.479 \pm 0.012$ | $0.502 \pm 0.016$ | $0.496 \pm 0.009$ | $0.517 \pm 0.012$ |
| Ours (n=8) | $0.491 \pm 0.012$ | $0.484 \pm 0.016$ | $0.485 \pm 0.016$ | $0.486 \pm 0.012$ | $0.491 \pm 0.012$ |
| Last-iter average | $0.489 \pm 0.008$ | $0.485 \pm 0.008$ | $0.495 \pm 0.009$ | $0.500 \pm 0.007$ | $0.514 \pm 0.007$ |
| Overall average | $0.528 \pm 0.004$ | $0.521 \pm 0.004$ | $0.530 \pm 0.005$ | $0.534 \pm 0.003$ | $0.544 \pm 0.003$ |

Tab. 8: Average frequency of using the agent itself as opponent, in the Soccer and Gomoku experiments. The frequency is calculated by counting over all agents and iterations. The $\pm$ shows the *standard deviations* estimated by 3 runs with different random seeds.

| Method | Ours ($n = 2$) | Ours ($n = 4$) | Ours ($n = 6$) |
|---|---|---|---|
| Frequency of self (Soccer) | $0.4983 \pm 0.0085$ | $0.2533 \pm 0.0072$ | $0.1650 \pm 0.0082$ |
| Frequency of self (Gomoku) | $0.5063 \pm 0.0153$ | $0.2312 \pm 0.0111$ | $0.1549 \pm 0.0103$ |

not set simply as the population for the sake of the proof. Alternatively, we pose some assumptions. Setting them to the population as in the main text may approximately satisfy the assumptions.

---

**Algorithm 2:** Simplified perturbation-based self-play policy optimization of one agent.

---

**Input:** $\eta_k$: learning rates, $m_k$: sample size;
**Result:** Pair of policies $(x, y)$;
1 Initialize $x^0, y^0$;
2 **for** $k = 0, 1, 2, \ldots \infty$ **do**
3     Construct candidate opponent sets $C_y^k$ and $C_x^k$;
4     Find perturbed $v^k = \arg\max_{y \in C_y^k} \hat{f}(x^k, y)$ and perturbed $u^k = \arg\min_{x \in C_x^k} \hat{f}(x, y^k)$ where the
        evaluation is done with Eq. 4 and sample size $m_k$ ;
5     Compute estimated duality gap $\hat{E}_k = \hat{f}(x^k, v^k) - \hat{f}(u^k, y^k)$;
6     **if** $\hat{E}_k \leq 3\epsilon$ **then**
7         **return** $(x^k, y^k)$
8     Estimate policy gradients $\hat{g}_x^k = \hat{\nabla}_x f(x^k, v^k)$ and $\hat{g}_y^k = \hat{\nabla}_y f(u^k, y^k)$ w/ Eq. 5 and sample size $m_k$;
9     Update policy parameters with $x^{k+1} \leftarrow x^k - \eta_k \hat{g}_x^k$ and $y^{k+1} \leftarrow y^k + \eta_k \hat{g}_y^k$;

---

## B.1 PROOF OF THEOREM 1

We restate the assumptions and the theorem here more clearly for reference.

**Assumption B.1.** $X, Y \subseteq \mathbb{R}^d$ $(d > 1)$ are compact sets. As a consequence, there exists $D \geq 1$, s.t.,

$$\forall x_1, x_2 \in X, \ \|x_1 - x_2\|_1 \leq D \ \text{ and } \ \forall y_1, y_2 \in Y, \ \|y_1 - y_2\|_1 \leq D.$$

Further, assume $f \colon X \times Y \mapsto \mathbb{R}$ is a bounded convex-concave function.

**Assumption B.2.** $C_y^k, C_x^k$ are compact subsets of $X$ and $Y$. Assume that a sequence $(x^k, y^k) \to (\hat{x}, \hat{y}) \wedge f(x^k, v^k) - f(u^k, y^k) \to 0$ for some $v^k \in C_y^k$ and $u^k \in C_x^k$ implies $(\hat{x}, \hat{y})$ is a saddle point.

**Theorem 1** (Convergence with exact gradients (Kallio & Ruszczynski, 1994)). *Under Assump. B.1,B.2, let the learning rate satisfies*

$$\eta_k < \frac{E_k}{\|g_x^k\|^2 + \|g_y^k\|^2},$$

*Alg. 2 (when replacing all estimates with true values) produces a sequence of points $\left\{(x^k, y^k)\right\}_{k=0}^{\infty}$ convergent to a saddle point.*

Assump. B.1 is standard, which is true if $f$ is based on a payoff table and $X, Y$ are probability simplex as in matrix games, or if $f$ is quadratic and $X, Y$ are unit-norm vectors. Assump. B.2 is about the regularity of the candidate opponent sets. This is true if $C_y^k, C_x^k$ are compact and $f(x^k, v^k) - f(u^k, y^k) = 0$ only at a saddle point $(u^k, v^k) \in C_y^k \times C_x^k$. An trivial example would be $C_x^k = X, C_y^k = Y$. Another example would be the proximal regions around $x^k, y^k$. In practice, Alg. 1 constructs the candidate sets from the population which needs to be adequately large and diverse to satisfy Assump. B.2 approximately.

The proof is due to (Kallio & Ruszczynski, 1994), which we paraphrase here.

*Proof.* We shall prove that one iteration of Alg. 2 decreases the distance between the current $(x^k, y^k)$ and the optimal $(x^*, y^*)$. Expand the squared distance,

$$\|x^{k+1} - x^*\|^2 \leq \|x^k + \eta_k g_x^k - x^*\|^2 = \|x^k - x^*\|^2 + 2\eta_k \langle g_x^k, x^k - x^* \rangle + \eta_k^2 \|g_x^k\|^2. \quad (6)$$

From Assump. B.1, convexity of $f(x, y)$ on $x$ gives

$$\langle g_x^k, x^k - x^* \rangle \geq f(x^k, v^k) - f(x^*, v^k) \quad (7)$$

which yields

$$\|x^{k+1} - x^*\|^2 \leq \|x^k - x^*\|^2 - 2\eta_k (f(x^k, v^k) - f(x^*, v^k)) + \eta_k^2 \|g_x^k\|^2. \quad (8)$$

Similarly for $y^k$, concavity of $f(x, y)$ on $y$ gives

$$\|y^{k+1} - y^*\|^2 \leq \|y^k - y^*\|^2 + 2\eta_k \left( f(u^k, y^k) - f(u^k, y^*) \right) + \eta_k^2 \|g_x^k\|^2. \quad (9)$$

Sum the two and notice the saddle point condition implies

$$f(x^*, v^k) \leq f(x^*, y^*) \leq f(u^k, y^*), \quad (10)$$

we have

$$\begin{aligned}
W_{k+1} &:= \|x^{k+1} - x^*\|^2 + \|y^{k+1} - y^*\|^2 \\
&\leq \|x^k - x^*\|^2 + \|y^k - y^*\|^2 \\
&\quad - 2\eta_k \left( f(x^k, v^k) - f(x^*, v^k) - f(u^k, y^k) + f(u^k, y^*) \right) + \eta_k^2 \left( \|g_x^k\|^2 + \|g_y^k\|^2 \right) \\
&\leq W_k - 2\eta_k E_k + \eta_k^2 \left( \|g_x^k\|^2 + \|g_y^k\|^2 \right).
\end{aligned} \quad (11)$$

If the learning rate satisfies $\eta_k < \frac{E_k}{\|g_x^k\|^2 + \|g_y^k\|^2}$, the sequence $\{W_k\}_{k=0}^{\infty}$ is strictly decreasing unless $E_k = 0$. Since $W_k$ is bounded below by 0, therefore $E_k \to 0$. Following from Assump. B.2, the convergent point $\lim_{k \to \infty} (x_k, y_k) = (x^*, y^*)$ is a saddle point. $\qquad\square$

## B.2 PROOF OF THEOREM 2

We restate the additional Assump. B.3 and the theorem here for reference. Assump. B.2 is replaced by the following approximated version B.4.

**Assumption B.3.** The total return is bounded by $R$, i.e., $|\sum_t \gamma^t r_t| \leq R$. The Q value estimator $\hat{Q}$ is unbiased and bounded by $R$ ($|\hat{Q}| \leq R$). And the policy has bounded gradient $\max\{\|\nabla \log \pi_\theta(a|s)\|_\infty, 1\} \leq B$ in terms of $L_\infty$ norm.

**Assumption B.4.** $C_y^k, C_x^k$ are compact subsets of $X$ and $Y$. Assume at iteration $k$, for some $(\hat{x}, \hat{y}) \in X \times Y$,

$$\forall (u, v) \in C_x^k \times C_y^k, \ f(u, \hat{y}) - \epsilon \leq f(\hat{x}, \hat{y}) \leq f(\hat{x}, v) + \epsilon$$

implies

$$\forall (u, v) \in X \times Y, \quad f(u, \hat{y}) - \epsilon \leq f(\hat{x}, \hat{y}) \leq f(\hat{x}, v) + \epsilon,$$

namely, $(\hat{x}, \hat{y})$ is an $\epsilon$-approximate saddle point.

**Theorem 2** (Convergence with policy gradients). *Under Assump. B.1,B.3,B.4, let sample size at step $k$ be*

$$m_k \geq \frac{2R^2 B^2 D^2}{\epsilon^2} \log \frac{2d}{\delta 2^{-k}}$$

*and, with $0 \leq \alpha \leq 2$, let the learning rate*

$$\eta_k = \alpha \frac{\hat{E}_k - 2\epsilon}{\|\hat{g}_x^k\|^2 + \|\hat{g}_y^k\|^2}.$$

*Then with probability at least $1 - \mathcal{O}(\delta)$, the Monte Carlo version of Alg. 2 generates a sequence of points $\{(x^k, y^k)\}_{k=0}^\infty$ convergent to an $\mathcal{O}(\epsilon)$-approximate equilibrium $(\bar{x}, \bar{y})$. That is*

$$\forall x \in X, \forall y \in Y, \quad f(x, \bar{y}) - \mathcal{O}(\epsilon) \leq f(\bar{x}, \bar{y}) \leq f(\bar{x}, y) + \mathcal{O}(\epsilon).$$

In the stochastic game (or reinforcement learning) setting, we construct estimates for $f(x, y)$ (Eq. 4) and policy gradients $\nabla_x f, \nabla_y f$ (Eq. 5) with samples. Intuitively speaking, when the samples are large enough, we can bound the deviation between the true values and the estimates by concentration inequalities, then the similar proof outline also goes through.

Let us first define the concept of $\epsilon$-subgradient for convex functions and $\epsilon$-supergradient for concave functions. Then we calculate how many samples are needed for accurate gradient estimation in Lemma 3 with high probability. With Lemma 3, we will be able to show that the Monte Carlo policy gradient estimates are good enough to be $\epsilon$-subgradients when sample size is large in Lemma 4.

**Definition 1.** An $\epsilon$-subgradient of a convex function $h : \mathbb{R}^d \mapsto \mathbb{R}$ at $x$ is $g \in \mathbb{R}^d$ that satisfies

$$\forall x', h(x') - h(x) \geq \langle g, x' - x \rangle - \epsilon.$$

Similarly, an $\epsilon$-supergradient of a concave function $h : \mathbb{R}^d \mapsto \mathbb{R}$ at $x$ is $g \in \mathbb{R}^d$ that satisfies

$$\forall x', h(x') - h(x) \leq \langle g, x' - x \rangle + \epsilon.$$

**Lemma 3** (Policy gradient sample size). *Consider $x$ or $y$ alone and treat the problem as MDP. Suppose Assump. B.3 is satisfied. Then with independently collected*

$$m \geq \frac{2R^2 B^2}{\epsilon^2} \log \frac{2d}{\delta}$$

*trajectories $\{(s_t^i, a_t^i, \hat{Q}_t^i)\}_{t=0}^T\}_{i=1}^m$, the policy gradient estimate*

$$\widehat{\nabla f} = \frac{1}{m} \sum_{i,t} \nabla \log \pi_\theta(a_t^i | s_t^i) \hat{Q}_t^i$$

*is $\epsilon$-close to the true gradient $\nabla f$ with high probability, namely,*

$$\Pr\left(\|\widehat{\nabla f} - \nabla f\|_\infty \leq \epsilon\right) \geq 1 - \delta.$$

*Proof.* It directly follows from Hoeffding's inequality and the union bound, since the range of each sample point is bounded by $RB$ and by the policy gradient theorem $\mathbb{E}\widehat{\nabla f} = \nabla f$. □

**Lemma 4** (Policy gradients are $\epsilon$ sub-/super- gradients). *Under Assump. B.1, the policy gradient estimate $\widehat{\nabla_x f}$ in Lemma 3 is an $\epsilon D$-subgradient of $f$ at $x$, i.e., for all $x' \in X$,*

$$f(x', y) - f(x, y) \geq \langle \widehat{\nabla_x f}, x' - x \rangle - \epsilon D$$

*with probability $\geq 1 - \delta$. (And $\widehat{\nabla_y f}$ is $\epsilon D$-super-gradient for $y$.)*

*Proof.* Apply the telescoping trick,

$$
\begin{aligned}
\langle \widehat{\nabla_x f}, x' - x \rangle &= \langle \widehat{\nabla_x f} - \nabla_x f + \nabla_x f, x' - x \rangle \\
&= \langle \nabla_x f, x' - x \rangle + \langle \widehat{\nabla_x f} - \nabla_x f, x' - x \rangle \\
&\geq f(x', y) - f(x, y) + \langle \widehat{\nabla_x f} - \nabla_x f, x' - x \rangle.
\end{aligned}
\tag{12}
$$

With the sample size in Lemma 3, we know it holds that $\max_i |\widehat{\nabla_x f} - \nabla_x f|_i \leq \epsilon$ with probability $\geq 1 - \delta$. Hence, by Holder's inequality, the last part satisfies

$$\langle \widehat{\nabla_x f} - \nabla_x f, x' - x \rangle \geq -\langle |\widehat{\nabla_x f} - \nabla_x f|, |x' - x| \rangle \geq -\|\widehat{\nabla_x f} - \nabla_x f\|_\infty \|x' - x\|_1 \geq -\epsilon D. \tag{13}$$

The proof of $\widehat{\nabla_y f}$ being $\epsilon D$-super-gradient for $y$ is similar, hence omitted. $\qquad\square$

Similarly for accurate function value evaluation, we have the following lemma on sample size, which directly follows from Hoeffding's inequality.

**Lemma 5** (Evaluation sample size). *Suppose Assump. B.3 holds. Then with independently collected $m \geq \frac{2R^2}{\epsilon^2} \log \frac{2}{\delta}$ trajectories $\{(s_t^i, a_t^i, r_t^i)\}_{t=0}^{T}\}_{i=1}^{m}$, the value estimate $\widehat{f} = \frac{1}{m} \sum_{i,t} \gamma^t r_t$ is $\epsilon$-close to the true gradient $f$ with high probability, namely, $\Pr\left(\|\widehat{f} - f\|_\infty \leq \epsilon\right) \geq 1 - \delta$.*

Now we prove our main theorem which guarantees the output of Alg. 2 is an approximate Nash with high probability. This is done by using Lemma 4 in place of the exact convexity condition to analyze the relationship between $W_k$ and $W_{k+1}$, using Lemma 5 to bound the error of policy evaluation, and analyzing the stop condition carefully.

*Proof.* (Theorem 2.)

Suppose $(x^*, y^*)$ is one saddle point of $f$. We shall prove that one iteration of Alg. 2 sufficiently decreases the squared distance between the current $(x^k, y^k)$ and $(x^*, y^*)$ defined as $W_k := \|x^k - x^*\|^2 + \|y^k - y^*\|^2$.

**Relation between $W_k$ and $W_{k+1}$:**
Note that

$$W_{k+1} = \|x^{k+1} - x^*\|^2 \leq \|x^k + \eta_k \hat{g}_x^k - x^*\|^2 = \|x^k - x^*\|^2 + 2\eta_k \langle \hat{g}_x^k, x^k - x^* \rangle + \eta_k^2 \|\hat{g}_x^k\|^2. \tag{14}$$

By Lemma 4, the gradient estimate $\hat{g}_x^k$ with sample size $m_k$ is an $\epsilon$-subgradient on $x$ with probability at least $1 - \delta/2^k$, i.e.,

$$\langle \hat{g}_x^k, x^k - x^* \rangle \geq f(x^k, v^k) - f(x^*, v^k) - \epsilon. \tag{15}$$

Plugging back into Eq. 14, we get

$$\|x^{k+1} - x^*\|^2 \leq \|x^k - x^*\|^2 - 2\eta_k \left(f(x^k, v^k) - f(x^*, v^k) - \epsilon\right) + \eta_k^2 \|g_x^k\|^2. \tag{16}$$

Similarly for $y^k$, since $\hat{g}_x^k$ is a super-gradient by Lemma 4,

$$\|y^{k+1} - y^*\|^2 \leq \|y^k - y^*\|^2 + 2\eta_k \left(f(u^k, y^k) - f(u^k, y^*) + \epsilon\right) + \eta_k^2 \|g_x^k\|^2. \tag{17}$$

Sum the two inequalities above, and notice the saddle point condition implies

$$f(x^*, v^k) \leq f(x^*, y^*) \leq f(u^k, y^*),$$

we have the following inequality holds with probability $1 - 2\delta/2^k$,

$$
\begin{aligned}
W_{k+1} = {}& \|x^{k+1} - x^*\|^2 + \|y^{k+1} - y^*\|^2 \\
\leq {}& \|x^k - x^*\|^2 + \|y^k - y^*\|^2 \\
& - 2\eta_k\big(f(x^k, v^k) - f(x^*, v^k) - f(u^k, y^k) + f(u^k, y^*) - 2\epsilon\big) + \eta_k^2\big(\|\hat{g}_x^k\|^2 + \|\hat{g}_y^k\|^2\big) \\
\leq {}& W_k - 2\eta_k(E_k - 2\epsilon) + \eta_k^2\big(\|\hat{g}_x^k\|^2 + \|\hat{g}_y^k\|^2\big).
\end{aligned}
\tag{18}
$$

**Accurate estimation of $E_k$:** In Eq. 18, the second term involves $E_k$ which is unknown to the algorithm. Recall that $E_k(u^k, v^k) = f(x^k, v^k) - f(u^k, y^k)$ and the empirical estimate $\hat{E}_k = \hat{f}(x^k, v^k) - \hat{f}(u^k, y^k)$ in Alg. 2 Line 5.

By Lemma 5, when the sample size $m_k$ is chosen as in Theorem 2, with probability $1 - \frac{2\delta}{d2^k}$,

$$
|\hat{f}(x^k, v^k) - f(x^k, v^k)| \leq \frac{\epsilon}{BD} \leq \epsilon
$$

and

$$
|\hat{f}(u^k, y^k) - f(u^k, y^k)| \leq \frac{\epsilon}{BD} \leq \epsilon.
$$

Thus $\hat{E}_k$ is $2\epsilon$-accurate because

$$
\begin{aligned}
\hat{E}_k - 2\epsilon = f(x^k, v^k) - \epsilon - f(u^k, y^k) - \epsilon &\leq E_k \\
&\leq f(x^k, v^k) + \epsilon - f(u^k, y^k) + \epsilon = \hat{E}_k + 2\epsilon.
\end{aligned}
\tag{19}
$$

**Case (1). Stop condition in Alg. 2 Line 6:** If there does not exist $(u, v) \in C_x^k \times C_y^k$ such that $\hat{E}_k(u, v) > 3\epsilon$, meaning $\forall (u, v) \in C_x^k \times C_y^k, \hat{E}_k \leq 3\epsilon$. We can conclude

$$
E_k = f(x^k, v) - f(u, y^k) \leq \hat{E}_k + 3\epsilon = 5\epsilon
\tag{20}
$$

with probability at least $1 - \frac{2\delta}{d2^k} \geq 1 - \frac{2\delta}{2^k}$.

Set $v = y^k$ and $u = x^k$ respectively in the above inequality, we obtain $\forall (u, v) \in C_x^k \times C_y^k$,

$$
f(u, y^k) - 5\epsilon \leq f(x^k, y^k) \leq f(x^k, v) + 5\epsilon.
\tag{21}
$$

Following from Assump. B.4, this implies $\forall (u, v) \in X \times Y, f(u, y^k) - 5\epsilon \leq f(x^k, y^k) \leq f(x^k, v) + 5\epsilon$, which suggests $(x^k, y^k)$ is an approximate saddle point (equilibrium).

On the other hand, we want to bound the failure probability. Define events

$$
F(g) := \text{``}|\hat{g} - g| \leq \epsilon \text{ is true''}
$$

for all $g \in \{g_x^0, g_y^0, f(x^0, v^0), f(u^0, y^0) \dots, g_y^k, f(x^k, y^k) \dots \}$. By De Morgan's law and the union bound,

$$
\begin{aligned}
&\Pr\big[\text{all MC estimates till step } k \text{ are } \epsilon \text{ accurate}\big] \\
&= \Pr\Big[\bigcap_{l=0}^{k} F(g_x^l) \cap F(g_y^l) \cap F(f(x^l, v^l)) \cap F(f(u^l, y^l))\Big] \\
&= 1 - \Pr\Big[\bigcup_{l=1}^{k} \overline{F(g_x^l) \cap \dots \cap F(f(u^l, y^l))}\Big] \\
&\geq 1 - \mathcal{O}\Big(\sum_{l=0}^{\infty} \frac{\delta}{2^l}\Big) \\
&\geq 1 - \mathcal{O}(\delta).
\end{aligned}
\tag{22}
$$

This means that inaccurate MC estimation (failure) occurs with small probability $\mathcal{O}(\delta)$. The purpose of the increasing $m_k$ w.r.t. $k$ is to handle the union bound and the geometric series here. So, when the algorithm stops, it returns $(\bar{x}, \bar{y}) = (x_k, y_k)$ as a $5\epsilon$-approximate solution to the saddle point (equilibrium) with high probability.

**Case (2). Sufficient decrease of $W_k$:** Otherwise, if the stop condition is not triggered, we have picked $u^k, v^k$ such that $\hat{E}_k > 3\epsilon$. With probability $1 - 2\delta$, $E_k > \hat{E}_k - 2\epsilon \geq \epsilon$. With the chosen learning rate $\eta_k$ in the theorem statement, $W_k$ strictly decreases by at least

$$W_k - W_{k+1} > \frac{\alpha(2-\alpha)\epsilon^2}{\|\hat{g}_x^k\|^2 + \|\hat{g}_y^k\|^2} \geq \frac{\alpha(2-\alpha)\epsilon^2}{2R^2B^2} > 0. \tag{23}$$

Since $W_k$ is bounded below by 0, by the monotone convergence theorem, there exists a finite $k$ such that $W_0 \leq k\frac{\alpha(2-\alpha)\epsilon^2}{2R^2B^2}$, and no $(u, v)$ can be found to decrease $W_k$ more than $3\epsilon$. In this case, $\forall (u, v) \in C_x^k \times C_y^k, \hat{E}_k(u, v) \leq 3\epsilon$, which is exactly the stop condition in Case (1). This means the algorithm will eventually stop, and the proof is complete. □

*Remark* 1. The sample size is chosen very loosely. More efficient ways to find perturbations (e.g., best-arm identification), to better characterize or cover the policy class and to better utilize trajectories (e.g., especially off-policy evaluation w/ importance sampling) can potentially reduce sample complexity. In practice, we found on-policy methods which do not reuse past experience such as A2C and PPO work well enough.

*Remark* 2. Assump. B.4 is a rather strong assumption on the candidate opponent sets. In theory, we can construct an $\epsilon$-covering of $f$ to satisfy the assumption. In practice, as in population-based training of Alg. 1, this assumption can be roughly met if $n$ is large or diverse enough. We found a relatively small population with randomly initialized agents already brought noticeable benefit.

*Remark* 3. The proof requires a variable learning rate $\eta_k$. However, the intuition is that the learning rate needs to be small, as we did in our experiments.

