# OpenReview forum: "Efficient Competitive Self-Play Policy Optimization"
_ICLR.cc/2021/Conference — Reject_

### Official Review · AnonReviewer4 · 2020-10-23
**A well-executed justification and evaluation of an intuitive opponent-selection rule**

**Rating:** 7
**Confidence:** 3

**Review:**

In this paper, the authors present a rule for selecting opponents for self-play training in zero-sum games: Train each agent i against the agent j that is "hardest" for i (in the sense that i's payoff is least among all candidate opponents when playing against j).  This principle is justified by appeal to "perturbation-based subgradient methods" in saddle-point optimization.  Last-iterate (as opposed to average-iterate) convergence to equilibrium is demonstrated both analytically and experimentally.

The evaluation is very thorough.  The empirical baselines are appropriately chosen.  The principled grounding of the selection procedure in saddle-point optimization was especially interesting, as I have not encountered that corner of optimization before.  The exposition is clearly written and well organized.

I have somewhat mixed feelings about this paper.  On the one hand, it's hard to believe that "train against the toughest opponent" as a selection rule has never been tried before.  On the other hand, the argument for the method is extremely convincing.  Overall I am in favor of acceptance.

It would be valuable for the authors to explicitly highlight the novel aspects of their contribution compared to [Balduzzi et al 2019], "Open-ended Learning in Symmetric Zero-sum Games", which rejects the "play against toughest (for me) opponent" approach explicitly because it collapses to equilibrium play.

---

### Official Review · AnonReviewer2 · 2020-10-27
**I acknowledge the paper and its results to be of interest for self-play in RL. However, in my opinion, the paper fails to properly account for the weakness of its approach. (I believe the paper to become stronger, if the efficiency of the approach was discussed critically.)**

**Rating:** 5
**Confidence:** 3

**Review:**

Summary:
The paper “Efficient Competitive Self-Play Policy Optimization” introduces a new self-play scheme for solving zero-sum two-player games. It is suggested to train a population of N agents in parallel, where each agent is matched against the comparatively strongest opponent in the next round of training. As baselines, the paper considers self-play against the best, the latest and random snapshots from the training history of only a single agent.

Strong:
The submission is well motivated and seems to state relevant related work.
The proposed algorithm seems straightforward and is presented in a clear and understandable way.
The proposed scheme solves for the Nash equilibrium for non-transitive matrix games with MC targets.
This is shown empirically and amended with a convergence proof.
The paper includes deep RL experiments for which the proposed algorithm results in better sample efficiency per agent.

Weak:
The last positive point also highlights the main weakness of the paper: I am not convinced, that the efficiency of RL self-play is best measured per agent. In the appendix, it is rightfully argued, that part of the training could be parallelized. However, the conclusion that the baseline experiments thus could be repeated N times, seems to ignore that the additional compute could be used to train stronger opponents also for the baselines. The experiments don’t account for this.
Further, the algorithm requires the evaluation of each agent-match-up combination for each round to choose the next opponent and thus involves policy roll-outs that are quadratic in the population size. This seems very expensive especially for larger populations. To highlight the efficiency of the approach in the title of the paper thus might be a misnomer.

Recommendation:
In its current form, I thus vote for a weak reject.

Support:
I acknowledge the paper and its results to be of interest for self-play in RL. However, in my opinion, the paper fails to properly account for the weakness of its approach. (I believe the paper to become stronger, if the efficiency of the approach was discussed critically.)

Rating: 5 out of 10

Confidence: 3 out of 5

CoE: I don’t see the paper in violation of the ICLR’s Code of Ethics.

---

### Official Review · AnonReviewer3 · 2020-10-28
**I highly encourage the authors to improve the clarity of their paper, and provide comparison with existing methods.**

**Rating:** 3
**Confidence:** 4

**Review:**

In this paper, the authors study the problem of optimization in two-player competitive zero-sum sequential games where the objective function is convex-concave in players' strategies. They relax the convex-concave assumption for the empirical study.

This problem is an important and challenging problem in the field of reinforcement learning.

The authors propose a policy gradient-based method and argue that it converges to the game's Nash equilibrium. They also conduct a series of empirical studies to show the significance of their method.

While this is a challenging problem, my judgment is that the paper is not ready for publication yet.

1) The presentation of theorem 1:
The authors state that if "a sequence of (x^k,y^k)->(xhat,yhat) \wedge f(x^k,y^k) -f(u^k,y^k)-> 0 implies (xhat,yhat) is a saddle point".
I am not sure I follow this sentence.
1-a) First, please clear up the notation when you use \wedge. For example, say ((x^k,y^k)->(xhat,yhat)) \wedge (f(x^k,y^k) -f(u^k,y^k)-> 0) since \wedge someitmes also means minimum.
1-b) In thm1, do the authors mean
if "a" sequence (x^k,y^k) for k>0, satisfies the following conditions ((x^k,y^k)->(xhat,yhat)) and (f(x^k,y^k) -f(u^k,y^k)-> 0), then (xhat,yhat) is a saddle point?
If that is the case, please consider rewording the thm1.
Also, if that is the case, for what \nu^k and u^k?

1-c) Algorithm 1 seems to produce a sequence of sets rather than (x^k,y^k). Please clarify it in your theorem.

1-d) If \nu^k and u^k are some quantities that are produced by your algorithm, then the theorem seems not to be well-stated. Because I am not sure what sequence of \nu^k and u^k the algorithm produces in order for me to see what space of X and Y you are dealing with.

1-e) I see the same lack of clarity in the proof of thm1.

2) In assumption 1, you state that the sets C_x^k and C_y^k are compact subsets of X and Y. However, these are what your algorithm produces. So probably, you may want to design your algorithm such it guarantees such a requirement.

3) E_k seems to be not defined.

As the authors stated, the main contribution of this paper is not the theory. And the theoretical study in this paper is to motivate their algorithm.
4) Empirical study.
I found the empirical study in this paper, somewhat not convincing.
If the contribution is empirical, it would be useful to compare it with existing methods.
I realized that the authors did not mention the work of
"Learning with Opponent-Learning Awareness" which proposes LOLA. Also, the authors mention the work of "Competitive policy optimization".
I strongly encourage the authors to make a clear empirical study and compare their method with prior works.

Since the proposed algorithm seems like a well-known traditional per-step best response method, such comparisons are helpful for future submission.

---

### Official Review · AnonReviewer1 · 2020-11-01
**Interesting paper**

**Rating:** 5
**Confidence:** 4

**Review:**


1. Summary:

This paper addressed an interesting topic on competitive self-play reinforcement learning on two-player zero-sum games. Typically, many self-play methods are only average-case optimal and self-play with latest agent fails to converge. I believe this is a big issue for many self-play because averaging past policies for each player during self-play iterations is very difficult especially for large-scale games.
The author tried to solve this problem by perturbation-based subgradient method because of its three advantages:
(1) it only requires to know the subgradients rather than the game dynamics;
(2) it's guaranteed to converge in its last iterate rather than the average iterate for convex-concave functions;
(3) it's very simple to select the opponent in the self-play training.
The authors evaluate this method on some small games and the experimental settings/results are well.
Overall, I like this paper. However, there are some issues or weakness in the current version, I tend to vote minor reject.
Note, if the author can address the key issues well, i can increase my score.

2. Some Concerns/weakness:

(1) For convex-concave functions, the perturbation-based subgradient method is guaranteed to converge in its last iterate. Does its convergence hold for deep learning settings?

(2) I'd like to see the performance on large-scale settings.
I think the strength of this method is to solve large-scale game, because it's expensive to average policies for the past iterations for many self-play methods. The author only evaluates the method on small settings, it's not very convincing because when solving small games, self-play with average policies is not expensive.

(3) The author missed some important related papers, such as neural counterfactual regret minimization [1, 2] and exploitability descent [3]. These methods also address the problem of average-case optimal. It will benefit readers if you can talk about these methods.

(4) the baseline is so weak, it will be better to compare against counterfactual regret minimization method and Johannes Heinrich's deep fictitious self-play method.

3. Questions:

(1) For large-scale games, does it need large population size $n$?

(2) Does it time-consuming to solve perturbed $v^{k}_i$ and $u^{k}_i$ for step 6 in Algorithm 1?

(3) Why not use exploitability to evaluate different methods? It's a standard metric in imperfect information games. It will be good if you can evaluate the method on one of poker game, such as leduc.

(4) "Four colors correspond to the 4 agents in the population with 4 initial points" in Figure 1, do you mean there are the population size is 4 and there are 4 different initial policies?

(5) what's exact gradient in "On the other hand, our method enjoys approximate last-iterate convergence with both exact and policy gradients."

(6) page 4, Algorithm 1, $N^0$ iterations, $N^0$ inner updates.  $N^0$ is a typo?


Some key references:

[1] Brown, N., Lerer, A., Gross, S. and Sandholm, T., 2019, May. Deep counterfactual regret minimization. In International conference on machine learning (pp. 793-802).

[2] Li, H., Hu, K., Zhang, S., Qi, Y. and Song, L., 2018. Double neural counterfactual regret minimization. ICLR, 2020. https://openreview.net/pdf?id=ByedzkrKvH

[3] Lockhart, E., Lanctot, M., Pérolat, J., Lespiau, J.B., Morrill, D., Timbers, F. and Tuyls, K., 2019. Computing approximate equilibria in sequential adversarial games by exploitability descent. arXiv preprint arXiv:1903.05614.

---

### Decision · Program_Chairs · 2021-01-07
**Final Decision**

**Decision:**

Reject

**Comment:**

This paper investigate an interesting problem of multi-agent RL with self-play. We agree with the reviewers that the paper requires more work before it can be presented at a top conference.  We would  encourage the authors to use the reviewers' feedback to improve the paper and resubmit to one of the upcoming conferences.